# DeepSlice: rapid fully automatic registration of mouse brain imaging to a volumetric atlas

Harry Carey[1,2], Michael Pegios[1], Lewis Martin[3], Chris Saleeba[1], Anita J. Turner [1], Nicholas A. Everett[4], Ingvild E. Bjerke [2], Maja A. Puchades [2], Jan G. Bjaalie [2] & Simon McMullan [1] ✉

Registration of data to a common frame of reference is an essential step in the analysis and integration of diverse neuroscientific data. To this end, volumetric brain atlases enable histological datasets to be spatially registered and analyzed, yet accurate registration remains expertise-dependent and slow. In order to address this limitation, we have trained a neural network, DeepSlice, to register mouse brain histological images to the Allen Brain Common Coordinate Framework, retaining registration accuracy while improving speed by >1000 fold.

A core concept in neuroscience is functional localization, in which different aspects of the brain's operation are mediated by physically distinct neural circuits. In this context, a necessary step in the analysis of neural data is their registration to the brain regions from which they were derived. For many years this need was met by paper-based stereotaxic atlases that provided standardization of nomenclature and defined regional boundaries in human and animal brains[1-6]. Recognizing the inherent limitations of two-dimensional atlases and inspired by the transformative influence of bioinformatics and information technology in the Human Genome Project[7], early brain imaging and neuroinformatics pioneers established volumetric atlas templates to assist with the registration of neural data[8-11].

For the mouse, the most used volumetric reference atlas is the Allen Mouse Brain Atlas[12], which contains delineations of hundreds of structures of the adult mouse brain within a standardized coordinate system, the Allen Common Coordinate Framework (CCF)[13], providing a foundation for the development of multimodal atlases that overlay gene expression, connectivity and function within the adult mouse brain[14-17].

Despite this progress and a pressing need for tools that facilitate standardization and integration of diverse data modalities[7,18-20], registration of images of histological sections to coordinate systems remains time-consuming and dependent on user skill. Some existing tools assume a knowledge of gross neuroanatomy[21], others assume programming proficiency[22-24], and can take hours to register multi-image datasets[22,24-26]. Recently, Convolutional Neural Networks (CNNs)

have shown great promise in the automated analysis of imaging data, including cellular histology[27,28] and pose estimation[29], but no equivalent tool for the registration of sectioned histological data to volumetric atlases has been described. Here we detail the development and performance of DeepSlice, a CNN trained on a massive histological dataset for the automatic alignment of coronal mouse brain histology to the CCFv3. DeepSlice is provided as an open-source Python package (github.com/PolarBean/DeepSlice) and an online tool (DeepSlice.org). Furthermore, it is compatible with the QuickNII registration tool and can be used as a first step in the QUINT workflow for the quantification of features in the rodent brain[30], enabling users to immediately begin using DeepSlice within a mature pipeline for quantifying brain data.

## Results

DeepSlice is based on the Xception CNN[31], modified such that the final layer has nine linear output neurons, regressing the corresponding CCF anchoring vectors ($O_{xyz}$, $U_{xyz}$, and $V_{xyz}$) used by the QuickNII histological alignment tool[21] (Fig. 1A). To train the model, we used a curated dataset of coronal mouse brain sections from the Allen Institute for Brain Science, which had already been aligned to the CCF. This included 131k images from the Allen database of slide-mounted histological sections processed for in situ hybridization (ISH), immunohistochemistry (IHC) or Nissl staining[14,32,33] (hereafter referred to as Allen histology), and 443k sections from the Allen Connectivity Atlas[16], which contained multichannel images of viral reporter expression captured using serial 2-photon block-face imaging (S2P, Fig. 1B). In the

[1]Macquarie Medical School, Faculty of Medicine, Health & Human Sciences, Macquarie University, Marsfield, NSW, Australia. [2]Department of Molecular Medicine, Institute of Basic Medical Sciences, University of Oslo, Oslo, Norway. [3]OpenBench, San Francisco, CA, USA. [4]School of Psychology, University of Sydney, Camperdown, NSW, Australia. ✉e-mail: simon.mcmullan@mq.edu.au

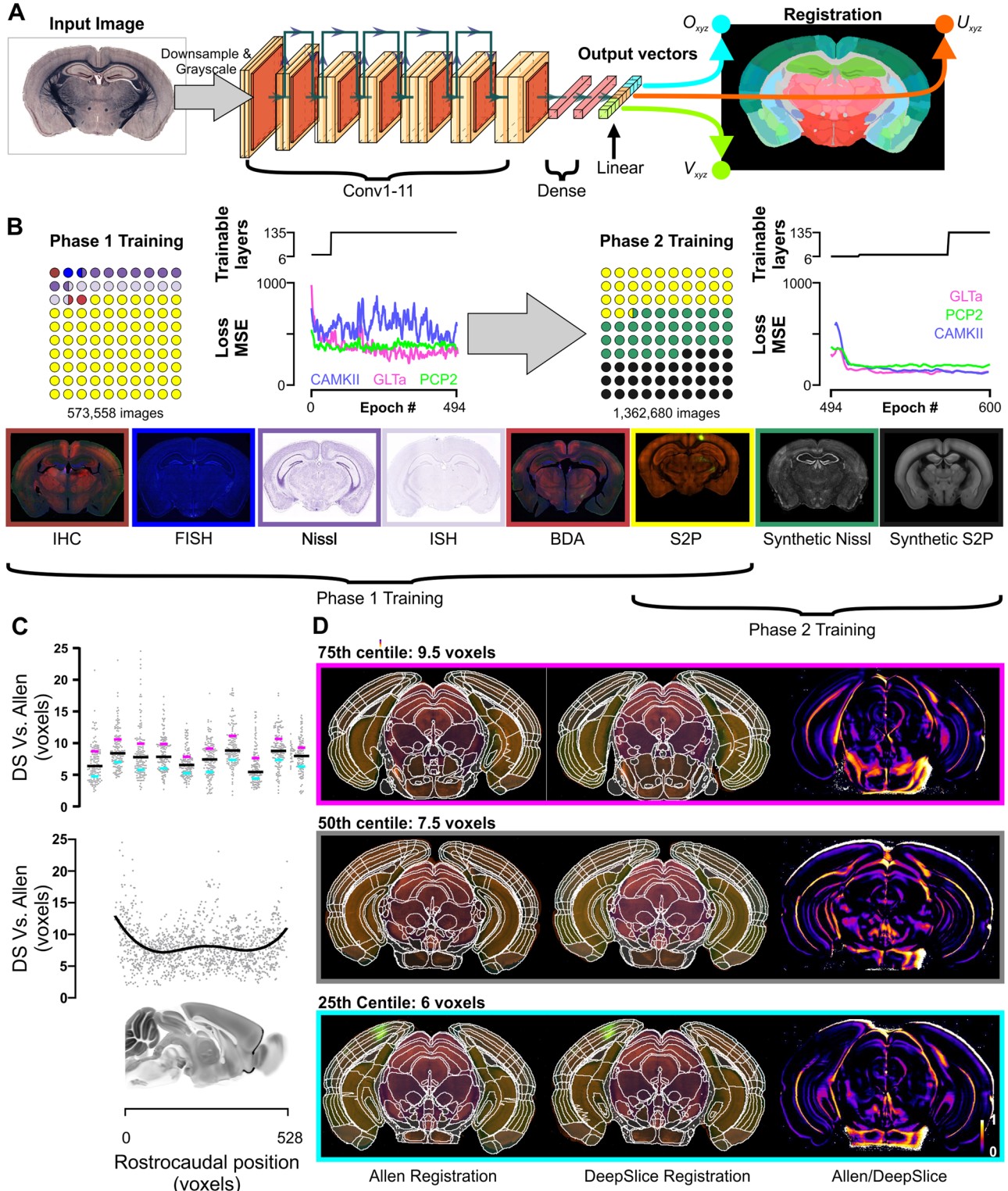

**Fig. 1 | DeepSlice Training and performance on serial 2-photon images.**
**A** DeepSlice (DS) is a convolutional neural network that analyses coronal mouse brain images and predicts output vectors ($O_{xyz}$, $U_{xyz}$, and $V_{xyz}$) corresponding to the coordinates of the image corners. **B** Training was conducted in two phases; the first used images of slide-mounted and serial 2-photon (S2P) sections, downloaded with their alignment metadata from the Allen Gene Expression (131k) and Connectivity Atlases (443k). The second phase used the same library of S2P images supplemented with a further 0.9 M synthetic coronal sections generated through the Allen Nissl and S2P volumes. Mean Squared Error (MSE) loss plots show performance against three human-aligned datasets of slide-mounted histology ($n = 119$ sections) with increasing rounds of training, before

(Phase 1) and after (Phase 2) addition of synthetic images. **C** Differences in predicted voxel positions between DeepSlice-and Allen-generated registrations in 1,400 sections from 10 S2P experiments that were omitted from the training set. Lines in the upper panel denote median and interquartile ranges; lower panel shows rostrocaudal distribution of error with a sagittal view of the mouse brain for reference. **D** Comparison of DeepSlice-generated registrations to corresponding Allen registrations; heat plots highlight differences between registrations. Examples shown are representative of 75th, 50th, and 25th centile performance across 10 S2P datasets. Atlas images in **A**–**C** courtesy of mouse.b-rain-map.org. Histological images in **D** from connectivity.brain-map.org experiment ID 509463587, 603717226, and 510582887.

following, we first describe the generation of Ground Truth alignments for slide-mounted sections, used for validation of the model. We go on to show how DeepSlice performed on these data and how performance was improved by (1) inclusion of a large set of synthetic images and (2) post-processing of DeepSlice-generated predictions. Lastly, we demonstrate the use of DeepSlice for brain-wide quantification of labeled neurons and show that DeepSlice gives very similar results to those acquired with human alignments.

## Benchmarking dataset

The data used for training DeepSlice did not provide optimal data for the validation of the model. On the one hand, the Allen histology dataset is large and representative of data acquired using different staining techniques. However, it was registered using automated methods, the fidelity of which has previously been reported as unreliable[34]. On the other hand, the S2P dataset had highly accurate registrations, but the uniform high quality of S2P images (obtained by incrementally scanning the exposed face of a volume of tissue during microtome sectioning[35]), is unrepresentative of slide-mounted histology, which is subject to tears, bubbles, and deformities. This lack of reliable Ground Truth registrations for slide-mounted sections initially made validation of model performance difficult. We therefore generated a 'Ground Truth' registration dataset of manually aligned slide-mounted histology datasets.

Human alignment of slide-mounted histology is inherently subjective, and thus inter-rater variability unavoidably occurs when human operators align the same images. Such inter-rater variability makes direct comparison of individuals difficult to interpret, as calculated differences quantify the disagreement between human operators without indicating which is superior. The same problem applies to comparison of DeepSlice to individual human operators: what is the appropriate 'Ground Truth' when even experts disagree?

One approach is to ask many operators to perform the same task and look at the pooled result, harnessing the "Wisdom of the Crowd" phenomenon in which the averaged output of a group may be more accurate than its most proficient individual[36,37]. We reasoned that, if human error is normally distributed, one might expect the averages of the human-assigned OUV anchoring coordinates to approximate the 'true' position of any section. We developed a benchmarking dataset consisting of 305 slide-mounted sections from 7 mouse brains (Supplementary Fig. 1), obtained from different laboratories and processed using common staining techniques. These sections were manually registered using QuickNII (RRID SCR_016854[21]) by 7 human operators of varying abilities (3 'novice' undergraduate students, 2 'intermediate' postgraduate/postdoctoral researchers, and 2 'expert' neuroanatomists with >10 years' experience). 'Ground Truth' registration coordinates for each image in the benchmarking dataset were calculated from the average of human-generated anchoring vectors (Supplementary Fig. 2). From this dataset, we randomly designated three brains as Validation ($n = 119$ sections; used to guide model development), and four as Test ($n = 191$ sections; used for final model assessment). The Ground Truth dataset, including images and human alignment coordinates, are available via the repository linked to this paper.

## Assessment of DeepSlice performance

When assessed against Validation data, the prototype model (trained only on Allen histology and S2P sections) generalized well to different staining and imaging modalities but had poor accuracy overall. We guessed that poor accuracy likely reflected contamination of the training dataset by misaligned sections, confirmed by *post hoc* analysis of original registrations (Supplementary Fig. 3). We found that such spurious training data could be identified by their high mean-squared error (MSE) values and excluded them from the image library used to train later models, but subsequent models still proved inaccurate

(Fig. 1B, Phase 1 Training). To improve the model further, we constructed a second training dataset of virtual tissue sections through the Allen Nissl and S2P template volumes[14,16] (Fig. 1B), based on real and stochastically generated anchoring coordinates. Real anchoring coordinates were harvested from metadata contained within 520,000 Allen histology and Allen Connectivity library images and used to generate synthetic images. A further 400,000 images were randomly generated from the Gaussian distributions of anchoring vectors contained within human-aligned Ground Truth datasets. This resulted in a large dataset of images that closely resembled the dimensions and cutting angles found within real-world slide-mounted sections (Supplementary Fig. 4). Since these images were generated from template volumes, each section corresponded exactly with its anchoring vectors, providing DeepSlice with Ground Truth training data. We enhanced variability of images appearance by random addition of noise, pixel drop-out, and warping (Supplementary Fig. 5). Since we found S2P registrations to be accurate, we included them in both phases of training. This approach yielded excellent generalizability and accuracy that extended to unseen imaging modalities (Fig. 1B, Phase 2 Training). Alignment vectors predicted by DeepSlice corresponded closely with vectors from ten unseen S2P datasets, quantified by estimating the average Euclidean distance (in 25 μm CCF voxels) between DeepSlice- and Allen-assigned atlas planes. For each dataset, the median distance between DeepSlice and Allen registrations was between 5.9 and 9.3 voxels (average $7.7 \pm 1.0$ voxels, -190 μm), with less agreement observed in sections closer to the rostral and caudal limits of the atlas (Fig. 1C, D).

## Post-processing

Despite these improvements, DeepSlice initially underperformed on slide-mounted Test sets, with a median registration error of $10.7 \pm 4.7$ voxels. This performance was enhanced by implementing post hoc adjustments to DeepSlice-generated predictions. First, since sequentially cut brain sections share the same cutting angles, we normalized the dorsoventral and mediolateral cutting angles of all sections within a dataset to equal the pooled average of DeepSlice-generated angle predictions, termed Angle Integration (AI: Fig. 2A). Second, we used cutting index metadata (CI), which are commonly saved in image filenames and denote the sequence in which sections are cut, to adjust rostrocaudal spacing between images (Fig. 2B). By measuring the accuracy of DeepSlice predictions in subsampled Test datasets of varying batch sizes, we determined that CI weighting enhanced prediction accuracy, particularly in datasets consisting of greater than 10-20 sections (Fig. 2C). Finally, vectors generated by the two DeepSlice models that performed best on the Validation sets were averaged (Model Ensembling, MEns).

Collectively, these modifications significantly improved DeepSlice performance on slide-mounted sections (Fig. 3). Predictions of dorsoventral and mediolateral cutting angles were highly correlated with Ground Truth measurements in S2P and human-aligned slide-mounted sections (Fig. 3B, $r^2 = 0.93$ and 0.97 respectively). Overall alignment accuracy was equivalent to human expert operators in unseen Test datasets (median error $6.6 \pm 1.7$ voxels, Fig. 3C, D), and performance on S2P data was improved by a further 25%, significantly reducing the difference between DeepSlice and Allen registrations to $5.8 \pm 1.4$ voxels (-140 μm, Tukey $P < 0.0001$).

## Test case: automated brain-wide quantification of neuronal labeling

To showcase the utility of DeepSlice and further establish its reliability, we used it to independently reanalyze human-aligned registrations which had previously been used to map the distribution of calbindin-immunoreactive neurons throughout the mouse brain[38,39] (Fig. 4). Images of histological sections and corresponding (human) alignment and annotation metadata were downloaded from the

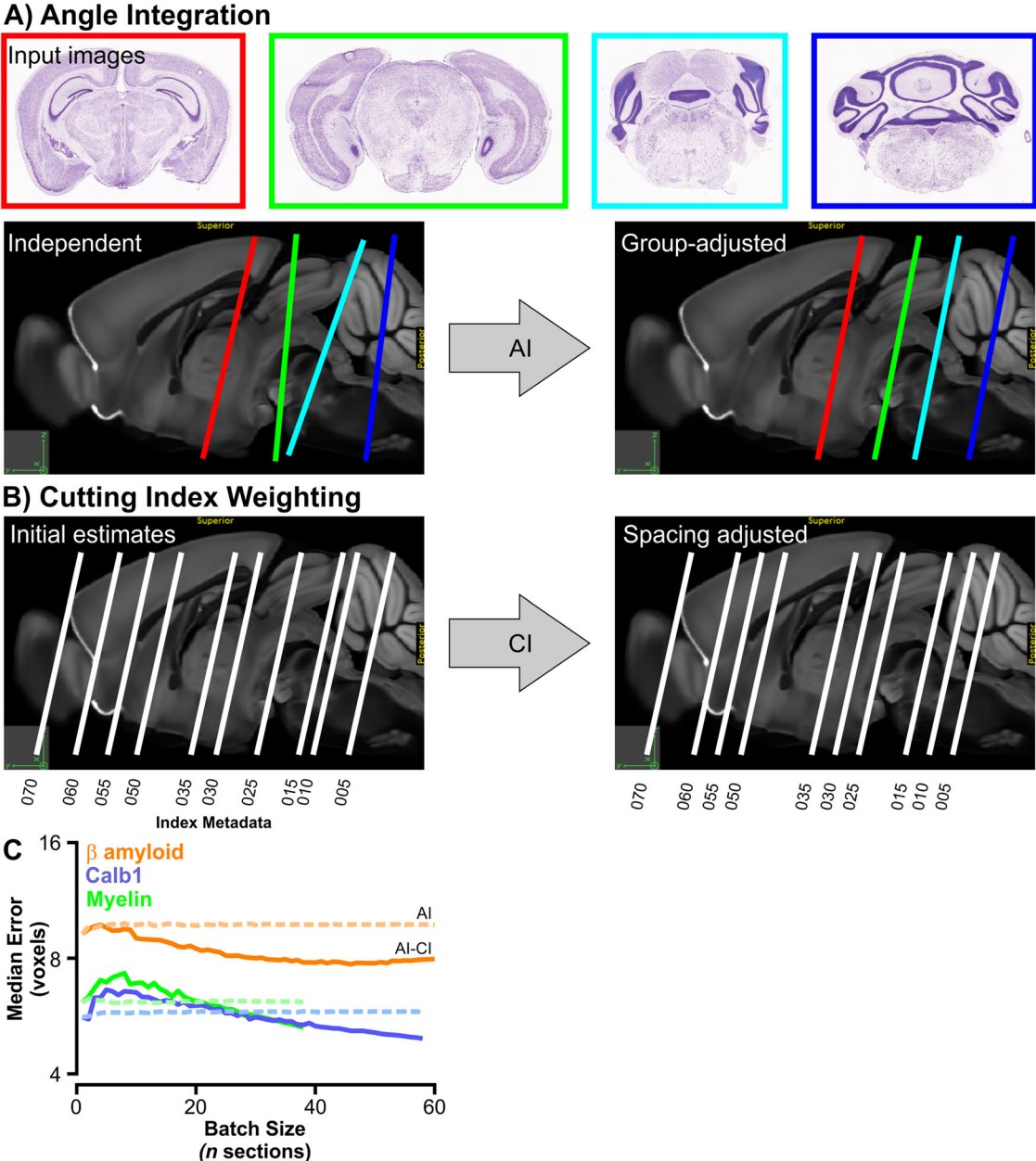

**Fig. 2 | *Post hoc* adjustments improved DeepSlice performance. A** Angle Integration (AI) normalized angles from individual predictions within a batch of images. **B** Cutting Index weighting (CI) uses section order metadata to predict the thickness of each section and uses this information to weight rostrocaudal spacing of sections. **C** The effect of increasing batch size on DeepSlice predictions before (dashed) and after (solid) CI weighting in three test datasets: Calb1: calbindin 1 in situ hybridization. Histology & atlas images courtesy of mouse.brain-map.org.

EBRAINS Knowledge Graph and aligned using DeepSlice. We then compared the regional distributions of calbindin-immunoreactive neurons resulting from human and DeepSlice registrations. We found that the distributions of neurons calculated from DeepSlice-generated alignment vectors was very similar to those calculated by an expert human operator ($R^2 > 0.97$ in all datasets, Supplementary Fig. 6B). This indicates that DeepSlice can produce reliable atlas registrations that can be integrated into existing quantification pipelines, which may be particularly useful for brain-wide gene/protein expression or connectivity studies.

## Discussion

Although a number of strategies have been developed for the registration of brain histology to reference atlases, to date these have required dedicated high powered computing infrastructure, extensive pre-processing of data, a high level of programming literacy and many hours of processing time per brain[22,24]. In contrast, DeepSlice is capable of generalizing across diverse imaging modalities and staining types and does not require pre-processing of images or specialized computer skills. DeepSlice is also computationally efficient, even on a mid-range 2019 laptop (74 ms/section with a Nvidia GTX 1060 GPU), typically aligning 50-section datasets in under 4 s compared to several hours taken by human operators in our study. Such simplicity and efficiency make DeepSlice deployable against enormous neuroimaging datasets. To increase the accessibility of DeepSlice, we have developed an online interface (www.DeepSlice.org) with drag-and-drop functionality, which produces CCF projection images and provides tabulated alignment vectors in QuickNII and VisuAlign (RRID:SCR_017978)-compatible formats. When using the web interface, users can elect for their images to be stored for use in the training

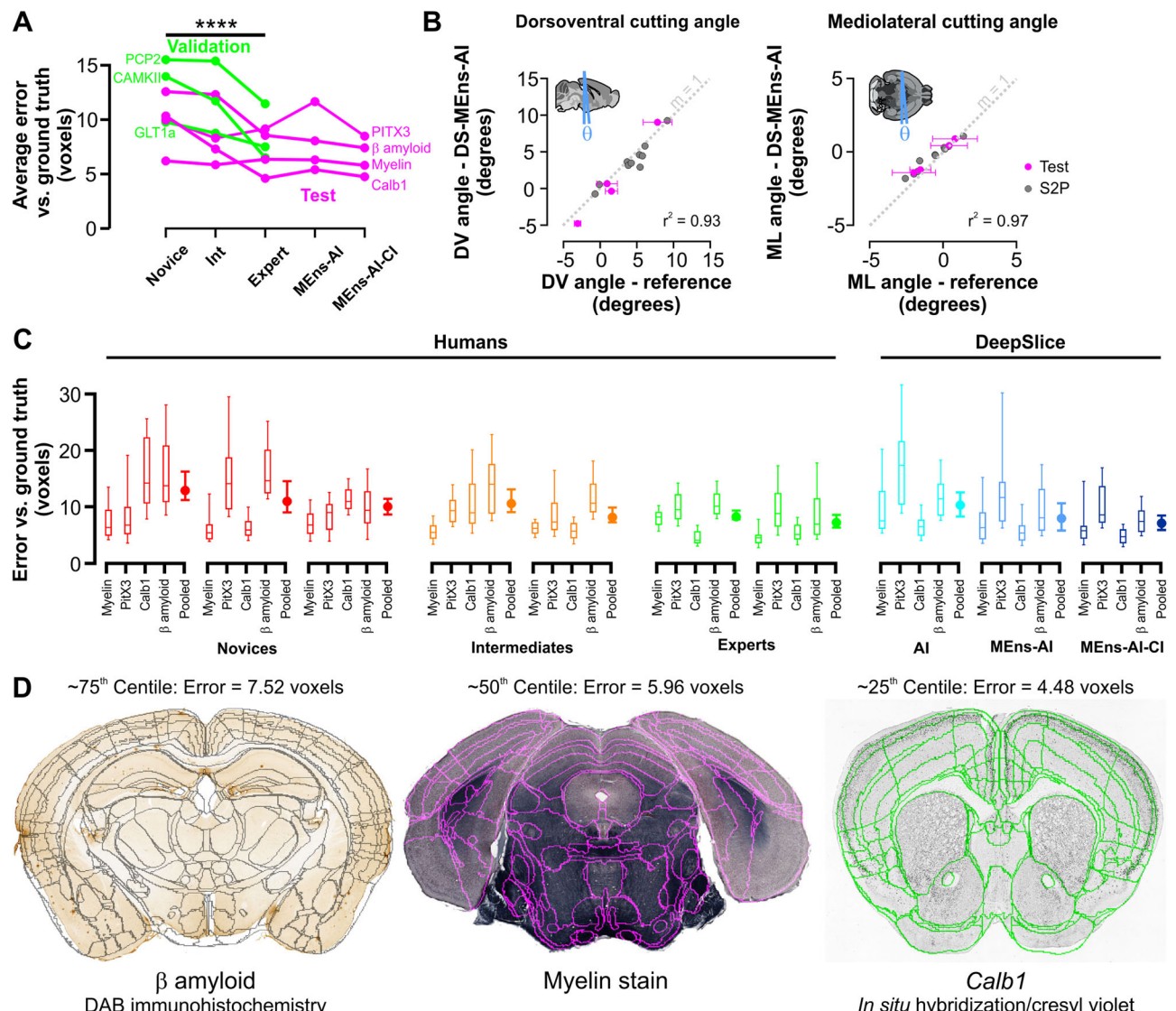

**Fig. 3 | Comparison of DeepSlice vs. human performance on slide-mounted coronal mouse-brain sections. A** Human alignment performance varied significantly by level of expertise on 7 datasets, which were randomly assigned as Validation (green), used for model refinement and development, and Test (magenta), used for assessment of final model performance. **** denotes a significance level of $P \leq 0.0001$, Tukey's repeated measures multiple comparison test. **B** DeepSlice Model Ensemble (DS-MEns)-predicted dorsoventral (DV, left scatterplot) and mediolateral (ML, right scatterplot) cutting angles corresponded closely with cutting angles in 'Ground Truth' S2P (gray, $n = 1400$ sections from 10 experiments) and human-aligned slide-mounted (magenta, $n = 191$ sections from 4 experiments) sections. Data are presented as mean values±SD. **C** Pooled data show

DeepSlice alignment accuracy with reference to human operators with varying levels of expertise; the ultimate DeepSlice iteration approximates human expert performance and consists of an ensembled model composed from the top performing 2 CNNs and integrated cutting index metadata (refer to Table 1 for details of test data). Boxes indicate 25–75th percentile range with median values indicated by horizontal lines; 10–90 percentile range is indicated by bars. Pooled data indicate median ± interquartile range **D**. Exemplars from the Test dataset (with overlaid alignment predictions) illustrating approximate 75th, 50th, and 25th centile performance of the MEns-AI-CI model. Schematic in **B** derived from S2P volume: mouse.brain-map.org. Panel **D** experiment details are provided in Table 1.

of future DeepSlice models, which will likely improve the robustness of DeepSlice performance on diverse data[40].

DeepSlice does have some weaknesses: like human aligners, DeepSlice relies on visual cues within sections to determine position. Thus, like human aligners, it tends to underperform on tissue in which neuroanatomical landmarks are obscured (e.g., low background staining, very high or very low signal contrast). Preparations that enhance background contrast (such as myelin stain) or closely resemble training data (Nissl and background fluorescence were highly represented) may improve reliability. Tissue deformity is also likely a source of error; whereas human operators may optimize alignment of deformed sections around the part of the brain that is most relevant to their work, DeepSlice has no such intuition and

appears to predict alignments in a more holistic manner. Further warping of training data may make future DeepSlice models more robust to section variability, but such efforts are unlikely to match those delivered by optimal sample preparation.

DeepSlice has been trained using images of whole coronal mouse brain sections, downsampled to a modest resolution, and can minimize errors in predicted dorsoventral and mediolateral cutting angles by integrating data from multiple images. For optimal results, we therefore make the following recommendations (see also Supplementary Fig. 6):

1. Use low-power images that capture the entire coronal section (rather than high-power images with a restricted field of view).

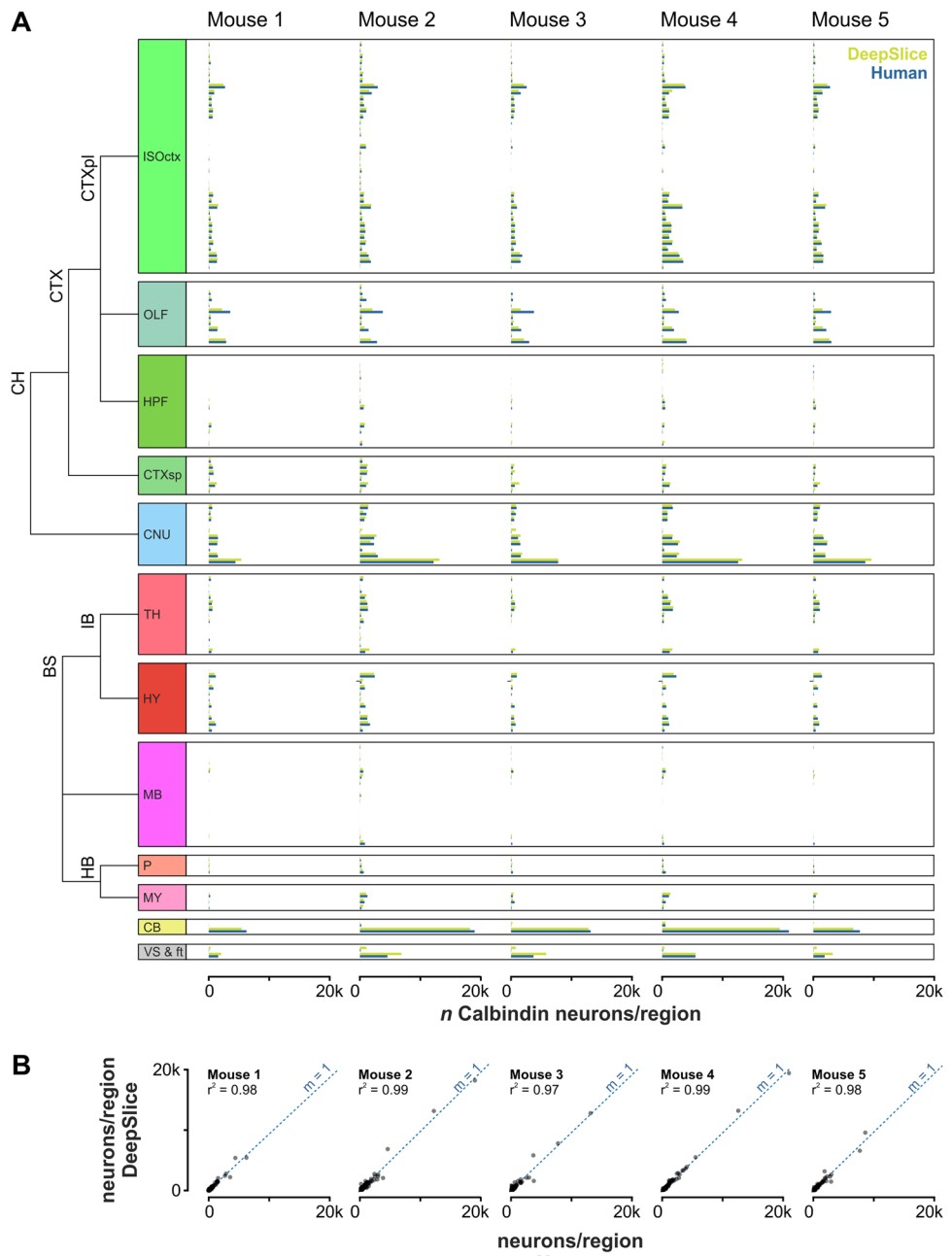

**Fig. 4 | A comparison of brain-wide regional cell counts for alignments generated by a human expert and by DeepSlice. A** Comparison of brain-wide regional cell counts following manual alignment of histological sections by a human operator and by DeepSlice; each bar denotes the number of Calbindin-immunoreactive neurons within subnuclei defined by the Allen Mouse Brain Atlas. **B** Linear regression of the same data. CH cerebrum, CTX cerebral cortex, CTXpl cortical plate, ISOctx isocortex, OLF olfactory areas, HPF hippocampal formation, CTXsp cortical subplate, CNU cerebral nuclei, BS brainstem, IB interbrain, TH thalamus, HY hypothalamus, MB midbrain, HB hindbrain, P pons, MY medulla, CB cerebellum, VS & ft ventricular systems & fiber tracts.

DeepSlice down-samples input images to 299 × 299 pixels, so there is no advantage to using larger image resolutions.

2.  Avoid using DeepSlice to align single images, even in cases where the region of interest is contained within one section (e.g., electrode track). Instead, users should group images obtained from a single experiment for batch analysis in order to benefit for the Angle Integration feature. Images do not need to be from adjacent sections to benefit from AI but do need to have been cut from the same tissue block. Ideally, users should note the order in which sections were cut and integrate this information into the image filename to benefit from *Cutting Index* weighting. For best results use at least 10-20 sections per batch.

3.  Adjust the brightness/contrast of images or consider assigning an additional image channel to ensure the whole section and anatomical landmarks within it are discernible; brightfield, DAPI, myelin stain, fluorescent NISSL and even unlabeled background fluorescence should work well. This may be particularly beneficial where the primary research data have a high signal-to-noise ratio (e.g., viral labeling). Merged multichannel images are also appropriate.

4.  Double-check DeepSlice alignments using the QuickNII tool and ensure that parts of the brain that are most important for your experiment are optimally aligned, adjusting in QuickNII as necessary. Compensate for any tissue warping or deformity using

tools such as VisuAlign (RRID: SCR_017978, https://www.nitrc.org/projects/visualign), which permit user-guided non-linear refinement of registrations.

In conclusion, DeepSlice aligns coronal mouse brain histology to the CCF in a reliable and rapid manner. In contrast to available registration methods[22–24]. DeepSlice is capable of generalizing across diverse imaging modalities and staining types, does not require preprocessing of images or specialized computer skills, and is deployable against large neuroimaging datasets. It simplifies the registration of mouse brain histology, and is integrated into mature analysis pipelines. It is also open-source, providing a foundation for further refinement by the community.

## Methods

### Model generation and training

Code was written in Python and the model constructed using Xception architecture, a pretrained CNN consisting of ~21 million trainable parameters that has high performance on benchmark ImageNet datasets, using the Keras machine learning library (https://keras.io/). The final Softmax layer of Xception was removed and replaced with two dense layers, each consisting of 256 neurons with rectified linear unit activation functions, and nine output neurons with linear activation functions corresponding to the $O_{xyz}$, $U_{xyz}$, and $V_{xyz}$ anchoring vectors used by QuickNII. Xception was initialized with weights pretrained on the ImageNet database. All input images were grayscaled and downsampled to $299 \times 299$ pixels to match the resolution of the images used to pretrain Xception, likely maximising the ability of the model to generalize.

Models were optimized using the mean squared error (MSE) loss function and the Adam optimizer[41], with an initial learning rate of 0.001 and batch size of eight. All Xception layers (except for batch normalization layers) were initially frozen with the only trainable layers being the final dense layers, which were randomly initialized. Layers were iteratively unfrozen as loss plateaued until the entire model was unfrozen (Fig. 1B); when the loss plateaued a final time the learning rate was further reduced to 0.0001. Pilot studies revealed that lowering the learning rate beyond 0.0001 yielded no further improvement. Performance against unseen holdout training images and human-aligned slide-mounted Validation sections was plotted every 5000 iterations. Training on an RTX 2080 Ti workstation took 3–4 days to reach convergence.

### Image library & curation

Pre-aligned coronal mouse brain images and corresponding alignment metadata were obtained via the Allen Brain Map application processing interface (http://api.brain-map.org). Training images included data from the Allen database of slide-mounted histological sections (including those stained for in-situ hybridization (ISH), immunohistochemistry (IHC), fluorescent in-situ hybridization (FISH), biotinylated dextran amine (BDA), and Nissl bodies), the Allen Connectivity atlas, and other S2P-acquired data. Alignment metadata were converted into $O_{xyz}$, $U_{xyz}$, and $V_{xyz}$ QuickNII anchoring vectors using the Allen2QuickNII script (https://github.com/Neural-Systems-at-UIO/allen2quicknii). During the assessment of prototype DeepSlice models we discovered that many training images returned unexpectedly high MSE values; in such cases, examination of the original data always revealed errors in the original alignment vectors (i.e., contamination of the training dataset: Supplementary Fig. 3). We therefore processed the entire training dataset through a prototype model and sorted images by MSE. For each data type (Nissl, ISH, IHC, FISH, BDA, etc.) we observed an inflection point at which alignment MSE grew exponentially (Supplementary Fig. 3A), which was used as a cut-off for elimination of potentially spurious training data. This resulted in a final curated library of 131,240 slide mounted sections (7,124 IHC, 7,181 FISH, 50,865 Nissl, 57,501 ISH, and 8,207 BDA).

No such outliers were identified in 442,680 images from the Allen Connectivity dataset, so all were included.

### Generation of synthetic datasets

To increase the number of training images, we generated a dataset of synthetic sections cut through the Allen Nissl and S2P reference volumes. To this end, we used the QSlicer tool (http://nesys.uio.no/QSlicer/QSlicer.zip) to generate virtual slices through each volume from sets of anchoring vectors. The list of input vectors was generated using real Allen alignment vectors and randomly generated vectors based on those found in the Ground Truth dataset. Synthetic images that contained minimal brain tissue (e.g., due to tangential cutting angles) were identifiable by small file size (<7 kb, $n = 58,000$), a result of image compression of their uniform black appearance, and excluded.

### Ground Truth datasets

Manual alignments of histological sections from 7 unseen experiments were used to assess DeepSlice performance (Table 1).

Ground Truth coordinates for each image were generated by averaging $O_{xyz}$, $U_{xyz}$, and $V_{xyz}$ coordinates assigned by seven human operators, classified as either Novice (<1 year of rodent neuroanatomy experience), Intermediate (2–10 years' experience), or Expert (>10 years' experience). Operators were provided with a two-hour QuickNII tutorial, after which they were provided access to test datasets. Over the following week operators began registration of their first dataset before a follow-up workshop that provided feedback and troubleshooting tips. Operators were allowed to complete remaining registrations at their own pace. Each operator was assigned seven datasets in total, the order of which was counterbalanced to mitigate practice effects.

### Data analysis

To quantify operator performance the alignment vectors generated by each were compared to the arithmetic mean of vectors generated by the other operators (i.e., for each image, $O_{xyz}$, $U_{xyz}$, and $V_{xyz}$ vectors generated by Operator 1 were compared to an ensembled alignment calculated by averaging vectors generated by Operators 2 – 7 etc., schematically illustrated in Supplementary Fig. 2). Differences between alignments were quantified by calculating the average Euclidean distance of pairs of voxels that correspond to the CCF-projected locations of each pixel in the image, masked such that voxels that fell outside the brain were excluded from analysis. The code used to quantify performance is included in the DeepSlice Github repository (https://github.com/PolarBean/DeepSlice/).

When assessing human performance on the 'Ground Truth' dataset, we found that accuracy was related to expertise, with novice operators performing worse than intermediate or expert operators (median error across seven datasets: $11.2 \pm 3.1$ vs. $10.3 \pm 3.8$ vs. $7.8 \pm 2.2$ voxels for novices, intermediates, and experts respectively, Tukey $P < 0.0001$ for all comparisons, Fig. 3A). Interestingly, the different datasets also represented a major source of variation in performance irrespective of experience level (2-way ANOVA $P = 0.0002$, $F_{(6, 28)} = 6.7$), matching operators' experience that some datasets were easier to align than others and suggesting that the Validation and Test sets were sufficiently diverse to generate meaningful data. Alignments generated by averaging vectors from all seven human operators were considered 'Ground Truth' and used for assessing DeepSlice performance.

Data were organized using Microsoft Excel or custom code generated using Python 3.7.7. Statistical analysis was performed using Graphpad Prism 9. The average performance of groups of human operators was compared using repeated measures ANOVA (Tukey).

**Table 1 | Datasets used to generate Ground Truth datasets**

| Name | # sections × section thickness | Description | Dataset DOI | Publication |
|---|---|---|---|---|
| GLT1a | 36 × 40 µm | Peroxidase-stained anti- glutamate transporter 1 (GLT1a) immunohistochemistry | 10.25493/Y147-2CE | Holmseth et al.[60] |
| PcP2 | 41 × 45 µm | X-gal treated Purkinje Cell Protein 2 (PCP2)-LacZ staining, fast red counterstain | 10.25493/A2EG-VPR | Lillehaug et al.[58] |
| CamKII | 47 × 25 µm | X-gal treated Calcium/calmodulin-dependent protein kinase (CamKII)-LacZ staining, Cresyl Violet counterstain | 10.25493/W97C-O1R | Odeh et al.[59] |
| Calb1 | 58 × 25 µm | Calbindin1 in situ hybridization | http://mouse.brain-map.org/experiment/show/71717640 | n/a |
| β amyloid | 60 × 40 µm | DAB-stained anti- β amyloid immunohistochemistry | n/a | Yates et al.[30] |
| Myelin | 41 × 50 µm | Myelin stain, sample collection 2565 | 10.25493/Y147-2CE | Holmseth et al.[60] |
| Pitx3 | 32 × 45 µm | X-gal treated Paired Like Homeodomain 3 (Pitx3)-LacZ staining, fast red counterstain | 10.25493/DTB3-0KP | Lillehaug et al.[58] |

Correspondence of cutting angles predicted by DeepSlice with dorsoventral and mediolateral cutting angles contained within S2P image metadata and average human-aligned angle estimates from the Test dataset was quantified by linear regression. $P < 0.05$ was considered statistically significant. Data were plotted using Graphpad Prism 9 or Matplotlib 3.5.3.

**Postprocessing of model predictions**
**Normalization of cutting angle.** Histological sections are iteratively cut in parallel sections from a single block of brain, which is difficult to align perfectly with orthogonal axes and therefore often contains imperfections in cutting angle. Whereas human experts integrate information contained within multiple sections to inform their estimates of dorsoventral and mediolateral cutting angle, DeepSlice analyzes each section independently, leading to section-to-section variability in the predictions of these values. Angle Integration (AI) measures cutting angles in every section from within an experiment, averages them, and then normalizes each individual alignment to match the average value of the group (Fig. 2A).

**Cutting index.** It is common practice for researchers to record the order in which histological sections are cut and to embed this information in the filenames of serial histological images (e.g., ExperimentID_s**001**.tiff, ExperimentID_s**002**.tiff, ExperimentID_s**003**.tiff etc.). The 'Cutting Index (CI)' can be used to estimate thickness of histological sections after initial alignment by DeepSlice, assuming constant section thickness within a dataset, and subsequently to adjust predictions of rostrocaudal position ($O_y$). Section thickness was calculated by dividing the difference in the value of $O_y$ between adjacent sections by the corresponding iterative change in cutting index and averaging this number across the entire dataset. To estimate the effect of batch size on CI-adjusted predictions we subsampled Test datasets in incrementally increasing batches up to 1000 times to account for the large number of possible permutations, and plotted median error of AI-adjusted predictions with and without CI weighting. The PITX3 dataset was omitted from this calculation because it had been prepared from two separate tissue blocks.

**Model ensembling.** We generated multiple DeepSlice models which, although trained using the same combination of images, differed subtly in the numbers of training iterations, level of noise, and the points at which layers were unlocked. Mirroring the 'Wisdom of the Crowd' approach used to establish Ground Truth for human-aligned slide-mounted histology, we also found benefits from averaging alignment vectors produced by the two best DeepSlice models (as determined by performance on the human-aligned Validation images), suggesting that sources of error differed in these models and canceled each other out. Ensembling more than two models was not found to significantly boost performance.

**Use case.** The QUINT workflow integrates the positions of features of interest within images (e.g., labeled neurons) with alignment metadata to generate corresponding CCF coordinates for each feature, and has been adopted as an approach for mapping viral labeling in connectivity studies[42–47], activity-dependent gene expression[48–51], and for defining regional boundaries of cell groups based on anatomical criteria[52,53]. To assess whether DeepSlice-generated alignments could reliably be used as part of this process, we directly compared regional distributions of calbindin-immunoreactive neurons in brains aligned by DeepSlice to those generated by an expert human operator. Images of coronal sections from five mice, corresponding alignment metadata (generated by a human operator using QuickNII) and annotation metadata (recording the positions of calbindin-positive neurons in each section, generated using ilastik[54]) were downloaded from the EBRAINS repository[39]. Annotation and alignment data were combined using

Nutil Quantifier 0.8.0 (RRID:SCR_017183[55]) using parameters for combining smaller CCF regions as defined by Bjerke et al.[38].

In the original study, calbindin-stained brains were cut transversely into two blocks to obtain a flat surface for microtome mounting and sectioning. Consequently, brain blocks were histologically sectioned in separate sessions and are assumed to have independent cutting angles. To account for this, sections from each block were processed separately by DeepSlice. Images were excluded if they did not include the whole section (hemi sections and sections with missing cortex), or if the operator had created separate alignments for a single image to account for significant displacement of section parts during slide mounting (e.g., in caudal regions where the cerebellum and brainstem are physically separated.

### Reporting summary

Further information on research design is available in the Nature Portfolio Reporting Summary linked to this article.

## Data availability

Histological images and human alignment data used to generate the Ground Truth library, and alignment predictions generated by the versions of DeepSlice discussed in the current paper, are available in Figshare with the identifier 10.25949/22802411. All other data are available at request from the authors. Source data are provided in this paper.

## Code availability

The data presented in the current paper were generated using Version 0.3 of DeepSlice, which is available at https://doi.org/10.5281/zenodo.8221471[56]. The most recent model of the DeepSlice Software package is available on PyPi (https://pypi.org/project/DeepSlice/) and github (10.5281/zenodo.8211292, https://github.com/PolarBean/DeepSlice[57]) under an open source license (GPL 3.0). A web application hosting DeepSlice is also available at https://www.deepslice.org.

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

## Acknowledgements

We are grateful to Ann Goodchild for her time-saving blunt assessments of many failed prototypes, for the motivation provided by Dr William Redmond, and especially to Veronica Downs, Freja Warner Van Dijk and Jayme McCutcheon, whose Novice alignments were instrumental to this work. We would like to thank Gergely Csúcs for providing his expertise and many atlasing tools. Work in the authors' laboratories is supported by the National Health & Medical Research Council of Australia, the Hillcrest Foundation, and Macquarie University (SMcM), and from the European Union's Horizon 2020 Framework Program for Research and Innovation under the Specific Grant Agreement No. 945539 (Human Brain Project SGA3) and the Research Council of Norway under Grant Agreement No. 269774 (INCF, JGB). We are grateful to Macquarie University for access to their HPC resources, essential for production of early DeepSlice prototypes.

## Author contributions

H.C. and S.M. conceived the study. H.C. and S.M. trained and tested machine learning models H.C. developed the Python package. M.P. and H.C. developed the web application. L.M. contributed code for angle adjustment which was integrated into the Python package. C.S., A.T., S.M., and M.A.P. assessed initial model performances and provided intermediate and expert registrations. I.E.B. created the brain-wide regional count comparison. J.G.P. and M.A.P. advised H.C. on the integration of DeepSlice with QuickNII. J.G.B. and N.E. provided conceptual input. Data analysis and statistics were performed by S.M. and H.C. The manuscript was written by H.C. and S.M. and edited by all authors.

## Competing interests

The authors declare no competing interests.
