## [Peer Review File · Nature Communications]

DeepSlice: rapid fully automatic registration of mouse brain imaging to a volumetric atlasREVIEWER COMMENTS

Reviewer #1 (Remarks to the Author):

DeepSlice: rapid fully automatic registration of mouse brain imaging to a volumetric atlas is an important advance in accurate image registration to an essential anatomic reference atlas. The registration of histology imaging, particularly partial volume and sectional data to standard reference atlases is a highly challenging problem and standard workflow of neuroanatomy. The Allen mouse brain atlas, as a common coordinate framework (CCF) has become the de facto standard for spatially resolved data analysis and presentation in mouse brain research. The authors point out that registration to CCFs is a laborious task with often several manual steps. Convolutional neural networks (CNN) offer strong potential for the basis for an effective image registration approach. DeepSlice provides powerful automated alignment of coronal slice histology and related images to the Allen CCF. The authors provide an open-source Python package for this work.

The author's approach is based on using the Xception CNN modified at the output layer so as to regress onto the necessary output alignment coordinates that the authors QuickNII tool requires for registration specification. The figures presented for comparison analysis are excellent and detail interesting and relevant comparison of issues and performance. The input training set is large consisting of 131k slide mounted histology and in situ hybridization and a more continuous set of brightfield and fluorescent image block sections that form the basis of the Allen CCF through two photon microscopy.

The authors correctly observe that there were substantial limitations in registration accuracy in the original Allen sections to the CCF. This occurred for a variety of reasons including poor section quality, more limited technology for alignment and other operational issues. The nature and methodology in the collection of images from the Allen Connectivity Atlas are far superior in quality and are used in constructing the CCFv3, so it is not surprising these would achieve accurate registrations.

The authors provide a rigorous set of benchmark tests for this work and the use of human image aligners provides for an interesting point of reference and benchmark. The authors also provide an honest assessment of the limitations of the approach as well.

I believe this to be an important advance in the use of neural net based approaches to mapping to common coordinate frameworks and can recommend publication.

Reviewer #2 (Remarks to the Author):

This work addresses the important problem of registration of sectional histological data to volumetric atlases. The method called DeepSlice is trained on a histological dataset for automatic alignment of coronal mouse brain histology to CCFv3. The network takes in a 2D histology section as input and outputs a set of coordinates to align with the 3D CCF.

Automatic registrations of the data are of poor quality as shown by manual inspection. S2P registrations are accurate but not representative of the data, which has additional artifacts such as tears and deformations. A benchmarking dataset containing 305 slide-mounted sections from 7 mouse brains is manually registered by 7 humans with varying skill.

To overcome poor quality of training data, 920K synthetic histological images were created from plausible anchor coordinates and sampled from AllenS2P and Nissl template volumes. Data augmentation is used to increase variability.

Strengths:

* Highly methodical work. Quantitative evaluation is performed by comparison of alignment vectors with vectors from ten unseen S2P datasets.

* High correlation between regional calbindin neurons quantified by DeepSlice versus expert demonstrates its efficacy. Supplementary figures are informative and provide a comprehensive analysis of performance.

Weaknesses:

* The image preprocessing and downsizing required in this work may lead to a further decrease in registration accuracy. One drawback of using deep learning models is that they cannot currently be applied to very high-resolution data. What are the effects of different amounts of downsizing on overall alignment performance?

* Grayscale all the different stains of histology may also lead to loss of anatomy/structure information (the staining is not simply for contrast, but to highlight different tissue/anatomy). What are the implications of learning different first few layers for different modalities to unify them into a canonical latent space?

Overall, this is strong work that provides a systematic approach to registering histological sections to the Allen CCF. The training setup, use of synthetic data and augmentations, and experimental validation are all well-justified.

Reviewer #3 (Remarks to the Author):

This is an interesting and compelling manuscript that describes the creation of software that aligns stained histological sections to a standard atlas. The software makes use of convolutional neural networks trained on both real and synthetic data to rapidly register novel images to the Allen Brain Atlas Common Coordinate Framework. Although this is by no means a new problem, it is a novel solution and a fast one at that.

The authors should acknowledge earlier work on the subject, including Mackenzie-Graham 2004 Journal of Anatomy, Martone 2004 Nature Neuroscience, Johnson 2007 NeuroImage, and Johnson 2010 NeuroImage. Also, credit where credit is due, Dr. Hong Wei Dong is the author of the Allen Brain Atlas his atlas should be cited, not just the CCF. It is interesting that the github refers to the Waxholm rat atlas (see Johnson above), but there is no mention of this in the manuscript. Perhaps space limitations of the journal? That said, the Deepslice.org website and github repository are excellent and the guide is very easy to read and understand.

The authors state that aligning a 50-section dataset takes 4 seconds, but few investigators collect 50 consecutive sections and stain them. Usually sections are collected in a region of interest and only a few of them are stained. How effective is DeepSlice in situations where there is sparse sampling? Perhaps only 2 or 3 slices. How about 1? Multiple sections can facilitate and make registration more robust, limiting the search space for the alignment. Also, how does DeepSlice handle high resolution imaging? Not only full slices, but smaller, higher magnification fields? Is there an option for an operator to manually "prime" a section by placing it in approximately the right location prior to automated alignment?

A minor issue is the numbering of the Supplementary Figures. The order makes no sense in the text of the manuscript, although the order does appear to make sense in the BioRxiv preprint, albeit with slightly different text in the preprint.

Also along these lines, the authors do not explicitly define a number of terms in the manuscript, such as PitX3, ISH, DAB, FISH, etc. Please spell all of these out in the manuscript. Further, exactly what stains used in the ISH and DAB is not mentioned in the text, but one can determine from Supp Fig 4. Please don't lead the reader on a treasure hunt to figure out what you mean in a manuscript, just say it up front.

Humans that align images are variously referred to as subjects, aligners, participants, and operators. Please choose one and stick to it, using different terms makes it hard to follow. Operators is my personal favorite, but I won't object if another term is used as long as it is consistent.

I believe that this work is a very interesting and exciting, well written, but perhaps a little too concise. I can't help but feel that the reason the writing is overly terse is because of a very limiting word count (1500 words). Nature Communications articles have a limit of 5000 words (not including abstract, Methods, References and figure legends). Please feel free to be more descriptive and maybe consider adding supplementary figure 6 as a regular figure (I feel that it is informative and should not be lost among supplementary figures).

REVIEWER 1

Key points:

- *DeepSlice: rapid fully automatic registration of mouse brain imaging to a volumetric atlas* is an important advance in accurate image registration to an essential anatomic reference atlas.
- DeepSlice provides powerful automated alignment of coronal slice histology and related images to the Allen CCF. The authors provide an open-source Python package for this work.
- The figures presented for comparison analysis are excellent and detail interesting and relevant comparison of issues and performance.
- The authors provide a rigorous set of benchmark tests for this work and the use of human image aligners provides for an interesting point of reference and benchmark.
- The authors also provide an honest assessment of the limitations of the approach as well.
- I believe this to be an important advance in the use of neural net based approaches to mapping to common coordinate frameworks and can recommend publication.

Author response: We thank Reviewer 1 for their kind words and generous assessment of our paper.

Action: None required

REVIEWER 2

Key points:

- Highly methodical work. Quantitative evaluation is performed by comparison of alignment vectors with vectors from ten unseen S2P datasets.

Author response: Thank you.

Action: None required

- High correlation between regional calbindin neurons quantified by DeepSlice versus expert demonstrates its efficacy. Supplementary figures are informative and provide a comprehensive analysis of performance.

Author response: Thank you.

Action: None required

- The image preprocessing and downsizing required in this work may lead to a further decrease in registration accuracy. One drawback of using deep learning models is that they cannot currently be applied to very high-resolution data. What are the effects of different amounts of downsizing on overall alignment performance?

Author response: This is a good question and one that we have pondered many times. We were initially concerned that the low image resolution used by Xception (299 x 299 pixels) may have been incompatible with high alignment performance, and so were pleasantly surprised when early prototypes revealed promising results. On reflection, our surprise was perhaps unfounded; down-sampled images are no harder to manually register than full resolution images (see Figure 1 for an example). This resolution is also quite similar to the size of the Allen STP and Nissl reference volumes, which measure ~450 x 320 voxels in the coronal plane. We conclude that, perhaps counterintuitively, 299 x 299 pixels is adequate to capture features necessary for accurate image registration by humans or machines.

Figure 1 Example of a coronal image through the medulla at full (LHS) and down-sampled resolution (RHS) – individual cells appear pixelated in the down-sampled version but gross anatomical features are easily discernable.

Action: The question “*What are the effects of different amounts of downsizing on overall alignment performance?*” is reasonable but difficult to determine empirically, as adjusting this would necessitate retraining DeepSlice from the ground up on the new input size. Furthermore, the underlying Xception model has also been pretrained on 299x299 images, and so changing this resolution may actually worsen the pretrained model’s performance too. To clarify this point we added this sentence at line 253 of the manuscript:

“All input images were grayscale and downsampled to 299 x 299 pixels to match the resolution of the images used to pretrain Xception, likely maximising the ability of the model to generalize”

And a sentence at line 212 where we make suggestions for optimal usage:

“Use low-power images that capture the entire coronal section (rather than high-power images with a restricted field of view). DeepSlice down-samples input images to 299 x 299 pixels, so there is no advantage to using larger image resolutions.”

- Grayscale all the different stains of histology may also lead to loss of anatomy/structure information (the staining is not simply for contrast, but to highlight different tissue/anatomy). What are the implications of learning different first few layers for different modalities to unify them into a canonical latent space?

Author response: Although greyscaling is a common method for reducing image variability in machine learning, determining when to apply it is somewhat subjective. We reasoned that application of a monochrome filter would reduce the heterogeneity of images, making it easier for DeepSlice to learn common features in tissues stained and imaged using different techniques.

On the other hand, it is not hard to think of situations in which greyscaling could be detrimental. In the example below (Figure 2), the bright green signal obscures the underlying red signal after greyscale conversion. As the red signal contains neuroanatomical details that are not visible in the high-contrast green signal, this kind of merged greyscale image could potentially impair performance: for cases like this, it would be better to split the channels and align the red channel in isolation. Although such images were highly represented in training,

Figure 2 Data loss following conversion of a merged RGB image to greyscale: the high green signal at the site of viral injection (top right side of colour image) obscures details that may be present in the underlying red channel after greyscaling. Picture credit: Allen Institute for Brain Sciences.

and despite the inclusion of pixel dropout as a pre-processing filter, habituating DeepSlice to partial image occlusion, loss of such information may still be relevant to performance.

When we initially designed DeepSlice we thought carefully about these issues, and in particular the question of whether the *specific colours* of features are likely to impart important information. We concluded that absolute colour was relatively unimportant (i.e. red NISSL contains the same information as blue NISSL), and that a more important question is whether there was sufficient ‘background’ staining to clearly differentiate anatomical features such as fibre tracts and edges. We concluded that that greyscaling of images was logical.

Action: As illustrated in the example above, users’ selection of which data to feed into DeepSlice is important and perhaps not intuitive. Contrast-rich signals, such as high-quality immunohistochemistry or viral labelling, may actually impair DeepSlice performance. However, an easy workaround is to capture anatomical features in a separate channel and use that channel for alignment. In the revised manuscript we have clarified this potential confound, recommended specifically allocating a channel for acquiring neuroanatomical features, and included an additional Supplementary Figure (6) that illustrates this workflow. Thank you for raising this concern. A sentence was added at line 224 to address this point:

“Adjust brightness/contrast of images or consider assigning an additional image channel to ensure the whole section and anatomical landmarks within it are discernible; brightfield, DAPI, myelin stain, fluorescent NISSL and even unlabeled background fluorescence should work well. This may be particularly beneficial where the primary research data have a high signal-to-noise ratio (e.g. viral labelling). Merged multichannel images are also appropriate.”

- Overall, this is strong work that provides a systematic approach to registering histological sections to the Allen CCF. The training setup, use of synthetic data and augmentations, and experimental validation are all well-justified.

Author response: We thank Reviewer 2 for their positive assessment of our work and thought-provoking questions.

REVIEWER 3

Key points:

- This is an interesting and compelling manuscript that describes the creation of software that aligns stained histological sections to a standard atlas... Although this is by no means a new problem, it is a novel solution and a fast one at that.

Author response: Thank you.

Action: None required

- The authors should acknowledge earlier work on the subject, including Mackenzie-Graham 2004 Journal of Anatomy, Martone 2004 Nature Neuroscience, Johnson 2007 NeuroImage, and Johnson 2010 NeuroImage. Also, credit where credit is due, Dr. Hong Wei Dong is the author of the Allen Brain Atlas his atlas should be cited, not just the CCF.

Author response: We thank the Reviewer for bringing this omission to our attention; we agree that the development of the CCF and the Allen Brain Atlas was built on the shoulders of a broader consortium of investigators who were trying to achieve similar goals.

Action: We have amended the Introduction (Lines 27-38) to provide a broader historical context for this work which includes reference to the investigators listed above and others.

- It is interesting that the github refers to the Waxholm rat atlas (see Johnson above), but there is no mention of this in the manuscript. Perhaps space limitations of the journal? That said, the DeepSlice.org website and github repository are excellent and the guide is very easy to read and understand.

Author response: Thank you for bringing this to our attention – the Github and website will host future implementations of DeepSlice that will register rat brain sections and sagittal mouse brain sections. Both projects are at a preliminary stage and sit beyond the scope of the current manuscript (the website now also includes a beta rat brain aligner).

Action: Website/Github amended for clarity.

- The authors state that aligning a 50-section dataset takes 4 seconds, but few investigators collect 50 consecutive sections and stain them. Usually sections are collected in a region of interest and only a few of them are stained. How effective is DeepSlice in situations where there is sparse sampling? Perhaps only 2 or 3 slices. How about 1? Multiple sections can facilitate and make registration more robust, limiting the search space for the alignment.

Author response: Thank you for this thoughtful question – we had wondered about this but had not previously tested it, the task being too computationally daunting and the benefits of Angle Integration and Cutting Index being clear when applied to entire datasets. To address this question we have randomly subsampled incremental combinations of sections from three of our Test datasets and plotted error against batch size, resampling up to 1000 times per batch to

account for the large number of possible permutations (Figure 3, below). As expected, we find that the accuracy of CI-adjusted alignments increases with batch size. In these 3 datasets the added benefits provided by CI emerged in batches of >4, >20, and >24 sections, improving predictions by ~20% on average. However, the take-home message should probably be that DeepSlice with AI post-processing actually does very well even with very small batch sizes (and, in this sample at least, seemed to outperform the AI+CI combination when only a couple of sections were analyzed, though this is likely due to small samples of sections).

Figure 2C Effect of increasing batch size on alignment accuracy using Cutting Index (CI) weighting. Each datapoint represents the median value of up to 1000 randomly sampled combinations of sections; calculations using Angle Integration (AI, dashed lines) are compared to AI with CI adjustment (solid lines).

Action: We thank the Reviewer for this stimulating question – we have included these data in the paper (Figure 2C) and suggest that users integrate batches of at least 10-20 sections for best results. A sentence describing this finding was added at line 148 of the manuscript.

“By measuring the accuracy of DeepSlice predictions in subsampled Test datasets of varying batch sizes, we determined that CI weighting enhanced prediction accuracy, particularly in datasets consisting of greater than 10-20 sections (Figure 2C).”

- Also, how does DeepSlice handle high resolution imaging? Not only full slices, but smaller, higher magnification fields? Is there an option for an operator to manually "prime" a section by placing it in approximately the right location prior to automated alignment?

Author response: Although users can load images of any resolution, DeepSlice automatically downsamples to 299 x 299 before processing, so there is no added advantage to uploading large image sizes: we recommend that users down-sample their images and use compressed image formats (e.g. PNG) prior to upload to the web portal to reduce transfer speed. Because the QuickNII coordinate system is scale-independent, results obtained from alignment of down-sampled images apply equally to the corresponding full-scale images; users can analyse and annotate their full resolution images (manually or using automated cell-detection tools like iLastik), align down-sampled images with DeepSlice, and integrate annotation and alignment metadata using tools like Nutli without any extra steps.

To answer the second question, at present there is no way to prime DeepSlice to optimally align high-magnification images; DeepSlice has been trained using images of whole brain sections at modest resolution, and so is unlikely to recognize images that only capture part of a section. In cases where users intend to use high-powered microscope images to capture detail, they would also need to capture the section at lower power in order to use DeepSlice to its full

potential.

(This leaves the problem of aligning high- and low-power images of the same samples! In principle this can be achieved from the image metadata, which captures microscope stage positions during acquisition and is already a feature of some imaging systems (e.g. Zeiss Zen Connect), allowing overlay of multiple images of the same sample taken by different devices (e.g. high power confocal image overlaid on low power slidescanner image: <https://www.zeiss.com/microscopy/en/resources/insights-hub/manufacturing-assembly/visualize-images-and-data-in-context---zeiss-zen-connect.html>)). This approach could equally be used by DeepSlice users to convert high-power images into CCF coordinates, although at present is not supported.

Action: This is a helpful perspective which we did not anticipate – we have added an additional section to the Discussion at line 212 which includes some recommendations on workflow optimization.

- A minor issue is the numbering of the Supplementary Figures. The order makes no sense in the text of the manuscript, although the order does appear to make sense in the BioRxiv preprint, albeit with slightly different text in the preprint.

Author response: Thank you for picking this up – this error occurred during reformatting.

Action: Corrected in the text.

- Also along these lines, the authors do not explicitly define a number of terms in the manuscript, such as PitX3, ISH, DAB, FISH, etc. Please spell all of these out in the manuscript. Further, exactly what stains used in the ISH and DAB is not mentioned in the text, but one can determine from Supp Fig 4. Please don't lead the reader on a treasure hunt to figure out what you mean in a manuscript, just say it up front.

Author response: Thank you for pointing out this oversight – we agree that this information should be included in the text.

Action: These abbreviations have now been defined in the text and figure legends and a table describing these datasets, which includes links to those in public repositories, has been added to the Methods section. The table has been added to the Methods and the legend of Supplementary Figure 1 has been updated.

- Humans that align images are variously referred to as subjects, aligners, participants, and operators. Please choose one and stick to it, using different terms makes it hard to follow. Operators is my personal favorite, but I won't object if another term is used as long as it is consistent.

Author response: Agreed! Thank you for this helpful comment.

Action: We have amended the manuscript to use the term 'operators' throughout, except for two usages of the term human aligners where operators would not have been appropriate.

- I believe that this work is a very interesting and exciting, well written, but perhaps a little too concise. I can't help but feel that the reason the writing is overly terse is because of a very limiting word count (1500 words). Nature Communications articles have a limit of 5000 words (not including abstract, Methods, References and figure legends). Please feel free to be more descriptive and maybe consider adding supplementary figure 6 as a regular figure (I feel that it is informative and should not be lost among supplementary figures).

Author response: Thank you for this constructive criticism; we agree that some elements of the text would benefit from expansion.

Action: We have redrafted the manuscript, expanding the text and reformatting the figures, and have included Supplementary Figure 6 as a main figure in the revised version (it is now Figure 2). We have also expanded the Discussion, and added an additional Supplementary Figure that highlights some workflow optimizations for DeepSlice. We believe these changes have improved the paper – thank you for your help.

REVIEWERS' COMMENTS

Reviewer #2 (Remarks to the Author):

This revised submission adequately addresses the issues raised in prior review.

Reviewer #3 (Remarks to the Author):

The authors have addressed my comments and the resulting manuscript is very good. Excellent work.